# The Effect of Various Cementing Agents on Occlusal Discrepancy Using an Intra-Oral Scanner: An In Vivo Study

Ameer Biadsee *, Rana Yassin, Eran Dolev 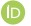, Vladimir Perlis, Shchada Masarwa and Zeev Ormianer

Department of Oral Rehabilitation, The Maurice and Gabriela Goldschleger School of Dental Medicine, Sackler Faculty of Medicine, Tel Aviv University, Tel Aviv 6997801, Israel; ranayassin@mail.tau.ac.il (R.Y.); eran@drdolev.com (E.D.); vovperlis@gmail.com (V.P.); masarwas@mail.tau.ac.il (S.M.); ormianer@tauex.tau.ac.il (Z.O.)
* Correspondence: ameerb@mail.tau.ac.il; Tel.: +972-525833906

**Abstract:** A marginal fit of all-ceramic crowns is a prerequisite for the long-term clinical success of a dental restoration. Few in vivo studies have investigated the effect of the film thickness of various luting agents on vertical discrepancy. This in vivo study evaluated the influence of three luting cements on the occlusal vertical discrepancy of milled crowns using a complete digital workflow. Forty-three patients treated in a students' clinic in Tel-Aviv University with 45 single posterior digitally prepared monolithic crowns were included in the study. The crowns were randomly divided into three groups using different resin luting agents: self-adhesive resin cement, resin-modified glass ionomer cement and adhesive resin cement. The crowns were intra-orally scanned before and after cementation. The two standard tessellation language (STL) files for each crown were superimposed using digital software, and between four and six measurements were made at the occlusal surface to demonstrate the occlusal and marginal discrepancies. One-way ANOVA ($\alpha = 0.05$) was used. The vertical occlusal discrepancy ranged from 2 to 38 µm. The mean vertical discrepancy values were (µm): self-adhesive resin = $12.93 \pm 4.74$, resin-modified glass ionomer = $19.05 \pm 4.60$ and adhesive resin = $13.69 \pm 5.17$. There were significant differences between resin-modified and self-adhesive cement groups ($p = 0.004$), and between resin-modified and adhesive resin cement groups ($p = 0.013$). Distal marginal ridge measurements were significantly different between resin-modified glass ionomer cement and self-adhesive resin cement group ($p < 0.001$) and the adhesive resin cement group ($p = 0.021$). There were no significant differences between the discrepancy values at the two measurement points in the self-adhesive cement group ($p = 0.377$), nor the resin-modified glass ionomer group ($p = 0.388$), or the adhesive resin cement group ($p = 0.905$). The cementation procedure with various resin cements results in occlusal vertical discrepancies within standard clinical acceptability. Resin-modified glass ionomer cement produced more vertical discrepancy than adhesive and self-adhesive resin cements did.

**Keywords:** dental cements; marginal misfit; CAD-CAM; occlusal discrepancy





## 1. Introduction

A marginal fit of all-ceramic crowns is considered a prerequisite for the long-term clinical success of dental restorations [1]. A complex interaction between variables related to dental restoration, luting agent and tooth structure may influence marginal misfit and microleakage [2]. A crucial factor affecting marginal discrepancy is dental cements, which act as barriers against microbial leakage, sealing the interface between the tooth and the restoration and holding them together through mechanical and/or chemical attachments [3]. Although marginal adaptation is a fundamental factor in fixed prostheses, there are limitations in analyzing its characteristics, especially with computer-aided design–computer-aided manufacturing (CAD-CAM) and all-ceramic restorations.

Terminology suggested by Holmes et al. defined the gap between prosthetic restoration and the prepared tooth as the marginal gap [4]. When no overextension or underextension is present, the marginal gap is equal to the absolute marginal discrepancy. However, recent studies have included terminology reporting marginal gap, marginal discrepancy, marginal fit and marginal adaptation [5–8].

There is no consensus on what constitutes a clinically acceptable maximum marginal gap. Numerous studies have adopted the criteria established by McLean and von Fraunhofer, who concluded that 120 µm was the maximum tolerable marginal gap [9].

The most common in vitro method of evaluating the marginal discrepancy was sectioning the cemented crowns and recording marginal measurements on digital photographs using a measuring microscope. In each study, different magnifications were used, and the results ranged from 116 to 165 µm [1,5–8]. On the other hand, Duati et al. reported mean marginal discrepancy values of 186 to 210 µm using micro-CT, without sectioning the cemented crowns. This nondestructive method allows 2D measurements of distances such as the absolute marginal discrepancy and internal gap, as well as the quantification of the cement space volume and the volume of porosities inside the luting agent. Consequently, the accuracy of the final restoration can be assessed after cementation, as well as between the stages of the working procedure [10]. Another in vitro study measured the distance between two predetermined points made on the prepared teeth and on the crowns before and after cementation using a profile projector under a torquing force. It found marginal discrepancies of 41 µm for resin cement and 49 µm for resin-modified glass ionomer cement. No statistical difference was found between groups [11]. An in vitro study by Rossetti et al. on Ni-Cr alloy crowns cemented using three different agents reported marginal gap values of 75.42 to 78.49 µm. Similar to the previous study, no significant difference was found between groups [1].

An in vitro study using a scanning electron microscope reported that resin cements exhibited a greater decrease in the marginal discrepancy than resin-modified glass ionomer cement following luting on an all-ceramic, complete veneer crown. The mean marginal gap after cementation was 116 µm for the resin group and 165 µm for the resin-modified glass ionomer group, with significant differences between groups [5].

Several in vivo studies evaluated the marginal fit of conventional cements by producing conventional impressions of the restoration before and after cementation using polyvinyl siloxane. A cast of each impression was poured using resin and was usually coated with gold. The replicas were viewed with a scanning electron microscope, and the marginal fit was examined, usually at 200× magnification. The marginal gap discrepancies were measured directly on the screen in 100 to 200 µm increments around the circumference of the tooth or by image analysis software and ranged up to $133.7 \pm 104.7$ µm [12,13].

Digital impressions made with an intra-oral scanner may be a way to improve the accuracy of dental restorations and research, as by their nature, these processes tend to eliminate the errors caused by conventional impression making and gypsum model casting [14]. Digital impression has high patient satisfaction and shorter clinical treatment time, and patients report fewer adverse condition such as gagging and suffocation hazards [14].

Only a few in vivo studies have investigated the marginal discrepancy of different luting agents and not using an intra-oral scanner [12,13]. Furthermore, all other destructive in vitro methods may induce a potential bias as a result of the fixation and sectioning process of the specimens. The purpose of this in vivo study was to determine and evaluate the effect on marginal fit using occlusal discrepancy measurements in an all-ceramic crown system cemented with three different luting agents using an intra-oral scanner. In a recent study investigating the precision of intra-oral scanners for single crowns, means varied from 11 to 60 µm [15]. A standardized unified scanning protocol can play an important role in the success of digital scanning [16].

The null hypothesis was that different cementation materials would not affect the vertical discrepancy.

## 2. Materials and Methods

This in vivo study was approved by the Tel-Aviv University Ethics Committee and was performed in the student dental clinic at Tel-Aviv University, School of Dental Medicine.

Inclusion criteria were periodontally healthy patients, fully dentate with stable occlusion, and a single posterior (molar or premolar) tooth intended to receive a monolithic crown adjacent to or between natural teeth. Exclusion criteria were patients with periodontal disease, absent adjacent tooth, subgingival preparations or practical difficulties in achieving accurate digital impressions.

The teeth were prepared to receive all ceramic crowns of lithium disilicate (IPS e.max CAD; Ivoclar Vivadent, Schaan, Liechtenstein). Preparations were completed using diamond burs F1R and F2R (Strauss Co., Raanana, Israel), with 1 mm chamfer finish line and occlusal reduction of 1.5 mm. The final preparations were checked by two clinical instructors using digital software (PrepCheck; Dentsply Sirona, York, PA, USA), which validated acceptable clinical preparation including convergence angle and equi-gingival finish line design. Digital impressions were made using an intra-oral, powder-free scanner (Omnicam; Dentsply Sirona). The crowns were digitally designed in CAD software (Cerec SW4.3; Dentsply Sirona) and milled (Cerec MC XL; Dentsply Sirona) with a spacer setting of 120 μm for the internal gap and 0 μm for the marginal gap. The designed occlusal scheme was designed according to the adjacent teeth. The milled crowns were then sintered and glazed following manufacturer's instructions.

The crowns were randomly divided into 3 groups and cemented to the prepared teeth using 3 different luting agents: (1) SARC group (n = 15) self-adhesive resin cement (G-CEM LinkAce, GC Corporate, Tokyo, Japan), (2) RMGIC group (n = 15) resin-modified glass ionomer cement (GC FujiCem2, GC Corporate), and (3) ARC group (n = 15) adhesive resin cement (G-CEM LinkForce, GC Corporate: Tokyo, Japan). Proximal inferences were examined, and digital bitewing X-rays were taken just before cementation using an XCP film holder (Dentsply) to ensure adequate fitting of the restoration.

The crowns were cemented following manufacturer's instructions. For etching, 5% hydrofluoric acid (Vita Ceramics Etch; Vita Zahnfabrik, Bad Säckingen, Germany) was applied for 20 s to the intaglio surface of the lithium disilicate crowns and rinsed with gentle water spray for 20 s. Subsequently, silane was applied on the inner surface (Monobond Plus; Ivoclar Vivadent).

In the SARC group, the resin material was placed in the crowns, which had been seated immediately using bite pressure for 3 s. Afterwards, each surface was tack cured with a visible light curing unit. The excess cement was removed, and each surface was cured again for 20 s.

In the RMGIC group, before applying the mixing tip, a small amount of paste was bled to ensure even flow from the cartridge. The cement was applied in the crowns and seated on the teeth. The excess cement was removed when it felt rubbery. Finger pressure was maintained for 5 min by a single operator (AB). In the ARC group, the teeth were etched with 37% phosphoric acid for 15 s, then gently rinsed and dried. An adhesive (G-Premio Bond; GC) was applied on the prepared surface. It was scrubbed for 10 s, dried for 5 s and then light cured for 10 s. The cement was applied to the internal surface of the crown with an automix syringe, and the crowns were seated on the respective teeth as described for the SARC group.

Intra-oral scans of the restoration including the two adjacent teeth using a standardized scanning protocol including a buccal starting point on the distal adjacent tooth to the buccal surface of the mesial adjacent tooth to the occlusal surfaces, finally to the lingual surfaces. All scans were performed by a single operator (AB) and were conducted at 2 time points: after final treatment of the crown immediately before cementation and after cementation after removing the excess cement. The scanning area was dried before each scan, and non-cemented crowns were seated for 5 min prior to scanning to ensure absence of gingival rebound. Each pair of standard tessellation language (STL) files (the initial and second scans) were imported into the inspection software (PolyWorks; InnovMetric,

Quebec, Canada) and superimposed using a best-fit alignment algorithm. To ensure precise superimposition, irrelevant areas such as below the cemento-enamel junction and beyond the field of interest were removed. Each pair of STL files were super-imposed twice by a single operator (SM) to ensure reliability and repeatability.

Six measurements at 6 different points were made in molars: mesio-lingual, disto-lingual, mesio-buccal, disto-buccal cusps, mesial and distal marginal ridges (Figure 1).

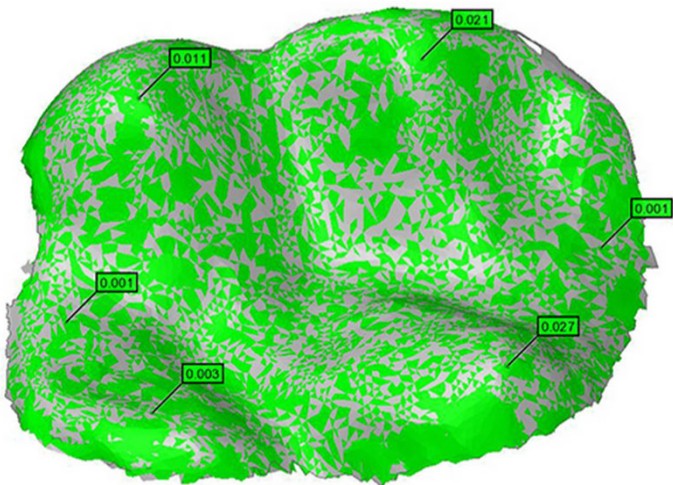

**Figure 1.** Six points showing differences (mm) between pre-cementation and post-cementation superimposed on standard tessellation language (STL) files showing 6 measurement points.

Due to the morphological differences between molars and premolars, 4 measurements were made of premolars, which included buccal and lingual cusps, mesial and distal marginal ridges. The measurements on the superimposed scans show the increased vertical occlusal discrepancy after cementation. Using the standardized scanning protocol, verified adequate crown fit was verified using bitewing X-rays. The vertical occlusal discrepancy represents the vertical marginal discrepancy.

*Statistical Analysis*

Data were analyzed using SPSS 25 for Windows (IBM Corp, Armonk, NY, USA). Descriptive analyses included mean, median and standard deviation. The Kruskal–Wallis test was performed to determine whether the occlusal discrepancies were significantly different between each group, followed by post hoc Bonferroni correction. Statistical significance was set at $p < 0.05$.

To test for a difference in means between 3 groups, according to a sample size calculation that considered test significance = 0.0166, power = 0.8, and effect size = 1.20, as was per formed using G*Power 3.1.9.4 software (Heinrich-Heine-Universität, Düsseldorf, Germany), a sample size = 14 in each group would be sufficient to achieve significant results that pass the Bonferroni corrections. The null hypothesis of the Friedman nonparametric test is that there are no differences between the variables. If the calculated probability is low ($p < 0.05$), the null hypothesis is rejected.

**3. Results**

This study included 43 patients (19 males and 24 females), ranging in age from 18 to 79 years. A total of 45 monolithic lithium disilicate all-ceramic crowns were produced, including 29 molars (64%) and 16 premolars (35%) were included, of which 28 (62%) were placed in the mandible and 17 (37%) in the maxilla. Among the 45 crowns, 15 were cemented using LinkAce resin cement, 15 with FujiCem2 cement, and 15 with LinkForce resin cement.

Mean vertical discrepancy values of the SARC group ranged from 2 to 38 μm. Overall mean vertical discrepancy was 12.93 ± 4.74 μm (Table 1).

**Table 1.** Occlusal vertical discrepancies (μm) of crowns in the self-adhesive resin cement (SARC) group measured at 6 points in each tooth.

| Tooth Area | N | Maximum | Minimum | Mean | Standard Deviation |
|---|---|---|---|---|---|
| Mesio-buccal | 15 | 38 | 3 | 16.26 | 9.96 |
| Distobuccal | 11 * | 23 | 6 | 14.64 | 5.22 |
| Distolingual | 15 | 34 | 2 | 14.27 | 8.92 |
| Mesio-lingual | 11 * | 23 | 6 | 14.64 | 4.76 |
| Mesial marginal ridge | 15 | 30 | 2 | 12.07 | 6.84 |
| Distal marginal ridge | 15 | 15 | 2 | 7.67 | 4.46 |
| Mean for tooth ** | 15 | | | 12.93 | 4.74 |

* Four pre-molars were included. The buccal cusp was considered as mesio-buccal cusp, and the lingual cusp as distolingual cusp. ** Overall mean of the 6/4 measurements per tooth.

Mean vertical discrepancy values in the RMGIC group ranged from 3 to 38 μm. Overall mean vertical discrepancy was $19.05 \pm 4.60$ μm (Table 2). Vertical discrepancy values in the ARC group ranged from 4 to 35 μm and the mean vertical discrepancy value was $13.69 \pm 5.17$ μm (Table 3). There were no significant differences between the discrepancy values at the 2 measurement points in the SARC group ($p = 0.377$), the RMGIC group ($p = 0.388$), or the ARC group ($p = 0.905$).

**Table 2.** Occlusal vertical discrepancies (μm) of crowns in resin-modified glass ionomer cement (RMGIC) group measured at 6 points in each tooth.

| Tooth Area | N | Maximum | Minimum | Mean | Standard Deviation |
|---|---|---|---|---|---|
| Mesio-buccal | 15 | 38 | 11 | 19.73 | 7.25 |
| Distobuccal | 10 * | 27 | 4 | 11.1 | 6.79 |
| Distolingual | 15 | 38 | 5 | 19.67 | 8.49 |
| Mesio-lingual | 10 * | 40 | 3 | 14.5 | 10.06 |
| Mesial marginal ridge | 15 | 70 | 3 | 16.8 | 15.59 |
| Distal marginal ridge | 15 | 50 | 10 | 25.13 | 13.28 |
| Mean for tooth ** | 15 | | | 19.05 | 4.60 |

* Five pre-molars were included. The buccal cusp was considered as mesio-buccal cusp, and the lingual cusp as distolingual cusp. ** Overall mean of the 6/4 measurements per tooth.

**Table 3.** Occlusal vertical discrepancies (μm) of crowns in adhesive resin cement (ARC) group measured at 6 points in each tooth.

| Tooth Area | N | Maximum | Minimum | Mean | Standard Deviation |
|---|---|---|---|---|---|
| Mesio-buccal | 15 | 35 | 12 | 14.34 | 9.06 |
| Distobuccal | 8 * | 32 | 1 | 12.5 | 9.11 |
| Distolingual | 15 | 33 | 6 | 14.47 | 8.07 |
| Mesio-lingual | 8 * | 23 | 8 | 15.38 | 5.55 |
| Mesial marginal ridge | 15 | 30 | 2 | 13.4 | 7.70 |
| Distal marginal ridge | 15 | 28 | 3 | 12.8 | 6.86 |
| Mean for tooth ** | 15 | | | 13.69 | 5.17 |

* Seven pre-molars were included. The buccal cusp was considered as mesio-buccal cusp, and the lingual cusp as distolingual cusp. ** The overall mean of the 6/4 measurements per tooth.

ANOVA with Bonferroni correction showed a significant difference in the mean vertical discrepancy values between groups. RMGIC group had significantly higher mean discrepancy values than the SARC ($p = 0.004$) and ARC groups ($p = 0.013$).

Distal marginal ridge (DMR) measurements were significantly different between the SARC-RMGIC ($p < 0.001$) and RMGIC-ARC ($p = 0.021$) groups.

## 4. Discussion

The current study used both CAD-CAM and a digital method to compare the marginal discrepancy before and after cementation. Several studies evaluating the marginal discrepancy of different luting agents have been published. There is no standardized method to measure the marginal fit, but the most common methods are the cross-sectional view, a direct view of the crown on a die, the impression replica technique and clinical examination [2]. The clinically acceptable marginal adaptation is controversial. However, McLean and von Fraunhofer suggested that a crown would be successful if the marginal gap was less than 120 μm [9].

Another result of the current study is that there was no statistically significant difference between the different measuring points among the three cement groups. However, RMGIC exhibited a greater difference in the distal marginal ridge (DMR) measurement point. This can be explained by differences in the thickness of each type of file: 10 μm for GC FujiCem2 cement, 4 μm for G-CEM LinkForce cement, and 3 μm G-CEM LinkAce cement. It is also strongly affected by the operator's manual skills, in addition to the non-uniform preparations of the teeth in in vivo studies.

During cementation, the cement space is filled. This is determined during the designing stage of CAD-CAM restorations. A study of marginal fit and internal adaptation of crowns found that a wider internal occlusal gap width favored the small marginal gap dimension [16].

A study on marginal and internal seals evaluated the adhesive interface of crowns using dye infiltration tests. They found that cement thickness did not influence the marginal seal [17]. The current study did not address this issue directly, but as shown in the results, RMGIC produced significantly more crown elevation, which may affect marginal seal.

The results of the current study showed a significant difference between the resin-modified glass ionomer cement group and the two resin cement groups. However, there was no significant difference between the self-adhesive resin cement group and the adhesive resin cement group. These data are supported by other studies. In an in vitro study on lithium disilicate crowns using a profile projector, Quintas et al. found a higher vertical marginal discrepancy of 49 μm for the resin-modified glass ionomer cement and 41 μm for the resin cement [11].

Similarly, Rosseti et al. reported a mean marginal gap of 78 μm for resin-modified glass ionomer cement and 74 μm for resin cement, with no statistical difference between groups [1]. Furthermore, Borges et al. measured a mean increase in vertical discrepancy of 54 μm after cementation for resin-modified glass ionomer cement and 36 μm for adhesive resin cements, with no significant difference [8]. Another in vitro study on sectioned zirconia crowns cemented to primary molar teeth used a digital stereomicroscope and reported a marginal gap of 140 μm for RMGI cement and 90 μm for the self-adhesive resin cement, with no statistical difference [7].

Other studies reported different marginal gap values. An in vitro study using a scanning electron microscope found that resin cements exhibited a 144 μm decrease in the marginal discrepancy and 105 μm for the resin-modified glass ionomer cement, following luting in all-ceramic complete veneer crowns, with significant statistical differences [5]. However, this study observed that the mean marginal discrepancy of all-ceramic restorations was about 200 μm prior to cementation, so the cement had enough space to be accommodated, facilitating better seating of the restoration. This may explain the greater marginal discrepancy values after cementation.

As previously reported, there was no significant difference between the adhesive and self-adhesive resin groups [10]. However, in an in vitro study, Duati et al. in, reported absolute marginal discrepancies of 186 to 195 μm for the resin cement groups. This is 10 times larger than those reported in the current study [10]. A possible explanation for these results might be that Duati et al., used identical replicated dies with accurate preparations, with 6 degrees convergence axial wall angle, which may affect the accuracy of the fitting. Moreover, micro-CT scans were used to assess the absolute marginal discrepancy.

The absolute results acquired from present study fall between the range of the scanner's deviation; however, using a strict standardized scanning protocol and a single operator may minimize the bias. In addition, results should be interpreted, not in their absolute values. There is an advantage to using a measurement software that accurate to the level discrepancy of a few microns between the two STL files. Although this was smaller than the range of the precision of the IOS, as reported, it may accentuate the difference between the two STL files.

The clinical importance of the results of this study is that RMGIC may increase the vertical marginal discrepancy when measuring the occlusal discrepancy when cementing lithium disilicate crowns.

This study had a few limitations. The sample was small. CAD/CAM manufacturing processes are variable, and the teeth were prepared by different operators. Clinical factors were not taken into consideration; for example, preparation convergence angle, height and finish line design were not evaluated. The patients' skeletal classifications were not considered in this study, although they may affect the cementation procedure. The in vivo scanning technique may affect the discrepancy results as well.

**5. Conclusions**

Under the limitations of this study, the cementation procedure with various resin cements resulted in occlusal vertical discrepancies that may affect the marginal vertical discrepancy within the standard range of clinical acceptability. Resin-modified glass ionomer cement exhibited higher occlusal vertical discrepancies than self-adhesive resin cement and adhesive resin cement.

**Author Contributions:** Conceptualization, A.B. and Z.O.; Methodology, A.B., E.D. and Z.O.; Investigation, A.B., R.Y., S.M. and V.P.; Resources, A.B., E.D. and V.P.; Writing, A.B. and R.Y.; Review and editing, Z.O. All authors have read and agreed to the published version of the manuscript.

**Funding:** This research did not receive external funding.

**Institutional Review Board Statement:** The study was conducted in accordance with the Declaration of Helsinki and approved by the Ethics Committee of Tel-Aviv University (21 September 2019).

**Informed Consent Statement:** Informed consent was obtained from all participants involved in the study.

**Data Availability Statement:** Data will be provided on request.

**Conflicts of Interest:** The authors declare that they have no conflict of interest.

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
