# Peer review of "The Effect of Various Cementing Agents on Occlusal Discrepancy Using an Intra-Oral Scanner: An In Vivo Study"

_applsci, doi:10.3390/app12126124_

Round 1

Reviewer 1 Report

Dear Authors, Dear Authors, first of all I would like to congratulate You on your work. The topic Is of great clinical relevance.
However, I believe that the article could be improved. Please, take a note of some suggestions.

Introduction- this section is short and should be more focused on the topic in question. If possible describe more details about intra oral scanner, 
I would suggest this paper "3D digital impression systems compared with traditional techniques in dentistry: A recent data systematic review
DOI10.3390/MA13081982. so you could add some informations about intra oral scanner. 

The use of the English language is reasonable, however, there are a number of punctuation and grammatical errors; that should be corrected and rephrased using academic English for a better flow of text for reader.

Abstract: is precisely written, and the aim of the study is mentioned. Please include some more information about the results/finding to enhance the impact of this section.

Furthermore, In discussion, more studies in context should be included; as there is little support of literature from the previous studies.

Author Response

Dear reviwer 1, 

Thank you for you valuable comment. 

Reviewer 2 Report

Dear authors,

this is an interesting manuscript about the influence of various cementing agents on the occlusal discrepancy obtained after the cementation in vivo.

The structure of the article, the architecture of the study and the statistics are well-structured. However, I suggest some specifications by the authors to clarify their work and improve the readability.

The type of occlusion in the patients treated is not specified (Angle class and skeletal class). Different pattern of occlusion may generate different levels of pressure on the tooth during the cementation phase. If it was not considered, please specify it in the limits of the study.

In addition, it is necessary to specify if the occlusion of the patients is stable; it is clear that there were no absences of adjacent teeth from the tooth prepared but it is not clear if the occlusion is stable for all the entire dental arch.   

It is not clear the CAD design of the dental crowns milled. Furthermore, the entity of the occlusal contacts established in the digital environment should be specified.

I counsel to reduce the introduction, focusing more on the statement of the problem that inspired the study.

Author Response

Dear reviewer 2, 

Thank you for you valuable comment. 
